# Regulation of Endothelium-Reticulum-Stress-Mediated Apoptotic Cell Death by a Polymethoxylated Flavone, Nobiletin, Through the Inhibition of Nuclear Translocation of Glyceraldehyde 3-Phosphate Dehydrogenase in Retinal Müller Cells

**DOI:** 10.3390/cells10030669

**Published:** 2021-03-17

**Authors:** Yoshiki Miyata, Kazuya Matsumoto, Shuichi Kusano, Yoshio Kusakabe, Yoshiya Katsura, Tetsuta Oshitari, Hiroshi Kosano

**Affiliations:** 1Faculty of Pharma-Sciences, Teikyo University, 2-11-1 Kaga, Itabashi-ku, Tokyo 173-8605, Japan; y-miyata@pharm.teikyo-u.ac.jp (Y.M.); matumoto@pharm.teikyo-u.ac.jp (K.M.); yoshio@pharm.teikyo-u.ac.jp (Y.K.); ostr@pharm.teikyo-u.ac.jp (T.O.); 2Fuji Sangyo Co., Ltd., 1301 Tamura-cho, Marugame, Kagawa 763-0071, Japan; s_kusano@fuji-sangyo.co.jp; 3The fifth Department of Internal Medicine, Tokyo Medical University, 3-20-1 Ami, Ibaraki 300-0332, Japan; y-katsu@tokyo-med.ac.jp

**Keywords:** nobiletin, Müller cell, ER stress, GAPDH, diabetic retinopathy, structure–activity relationship

## Abstract

In the early stages of diabetic retinopathy (DR), subtle biochemical and functional alterations occur in Müller cells, which are one of the components of the blood–retinal barrier (BRB). Müller cells are the principal glia of the retina and have shown a strong involvement in the maintenance of homeostasis and the development of retinal tissue. Their functional abnormalities and eventual loss have been correlated with a decrease in the tight junctions between endothelial cells and a consequent breakdown of the BRB, leading to the development of DR. We demonstrated that the endothelium reticulum (ER) triggers Müller cell death and that nuclear accumulation of glyceraldehyde 3-phosphate dehydrogenase is closely associated with ER-induced Müller cell death. In addition, induction of ER stress in Müller cells increased vascular endothelial growth factor expression but decreased pigment-epithelium-derived factor (PEDF) expression in Müller cells. We found that nobiletin, a polymethoxylated flavone from citrus explants, exerts protective action against ER-stress-induced Müller cell death. In addition, nobiletin was found to augment PEDF expression in Müller cells, which may lead to the protection of BRB integrity. These results suggest that nobiletin can be an attractive candidate for the protection of the BRB from breakdown in DR.

## 1. Introduction

Müller cells are one of glial cells in the retina and the signature cells to spread through the retina and are in contact with retinal vessels and neurons [1,2]. Müller cells possess critical functions in the modulation of blood flow and the maintenance of the blood–retinal barrier (BRB) [3,4,5]. The BRB is important for preventing the emigration of blood and other hazardous substances such as pathogens from the retinal tissue. There are several reports that Müller cells induce blood–retinal barrier integrity in retinal endothelial cells [2,6,7]. Studies using conditional ablation of Müller cells have shown severe BRB breakdown [2,8], but the mechanism of how Müller cells maintain the BRB is debated. Several reports suggest that Müller cells are involved in BRB properties by producing factors such as pigment-epithelium-derived factor (PEDF) and thrombospondin-1, which are anti-angiogenic and strengthen the endothelial barrier [4,9,10]. In contrast, the barrier function is impaired by vascular endothelial growth factor (VEGF) and matrix metalloproteinases (MMPs) derived from Müller cells [11] as MMPs lead to the degradation of the tight junction protein occuldin [12]. Abu El-Asrar et al. [13] demonstrated the relationship between vitreous levels of MMPs and VEGF in proliferative diabetic retinopathy (PDR). In addition, the hypoxic condition is known as one of the stimuli leading to the induction of these pathogenetic factors [14].

We have reported that MMP-9 activity and the secretion of tissue inhibitors of metalloproteinase, known as endogenic MMP inhibitors, are mediated by protein kinase C in a human Müller cell line (MIO-M1) [15]. In addition, as a consequence of screening for novel MMP inhibitors using natural and synthesized polymethoxylated flavones (PMFs) in MIO-M1 cells, we demonstrated that nobiletin, a polymethoxylated flavone from citrus explants, decreases MMP-9 activity through the inhibition of transcription for MMP-9 gene expression and increase in tissue inhibitor of metalloproteinase-1 secretion in Müller cells [16]. Furthermore, Müller cell death promotes the breakdown of BRB integrity, leading to increased vascular permeation and disappearance of neuroprotection, which affect sboth vascular cells and neurons [17]. Owing to their critical roles in maintaining retinal functions, any disturbance or cell damage to Müller cells is closely associated with the pathogenesis of various retinal diseases, including diabetic retinopathy (DR) [18]. Therefore, the protection of Müller cells from retinal pathogens may prove to be a promising therapy for DR.

Endothelium reticulum (ER) stress is a potential cause of retinal vascular and neuronal cell death in retinal diseases [19,20,21,22]. Wu et al. [23] have shown that oxidized glycated low-density lipoprotein (LDL) promotes ER stress and contributes to apoptotic cell death in Müller cells. However, the molecular mechanisms underpinning ER-stress-mediated apoptosis in Müller cells remain to be fully elucidated. In this study, we demonstrated that the nuclear translocation of glyceraldehyde 3-phosphate dehydrogenase (GAPDH) is involved in ER-stress-mediated apoptotic cell death in Müller cells. We found that nobiletin inhibits the nuclear translocation of GAPDH, which leads to the suppression of ER-stress-induced Müller cells. In addition, nobiletin may have protective action on the BRB structure through the augmentation of PEDF expression. These results suggest that nobiletin can be an attractive candidate for the protection of the BRB from breakdown in DR.

## 2. Materials and Methods

### 2.1. Materials

Small interfering RNAs (siRNAs) were synthesized by Thermo Fisher Scientific, Inc. (Waltham, MA). We obtained antibodies against C/EBP homologous protein (CHOP) (L63F7), BiP, and caspase-3 from Cell Signaling Technology, Inc. (Danvers, MA). We also obtained antibodies against GAPDH and beta-actin (β-actin), tunicamycin (Tm), thapsigargin (Tg), and other chemicals from Sigma-Aldrich (St. Louis, MO). Nobiletin (3′,4′,5,6,7,8-hexamethoxyflavone) was obtained from FUJIFILM Wako Pure Chemical Co. (Osaka, Japan). Following the previously reported protocol [24,25,26], we synthesized nobiletin congener.

### 2.2. Cell Culture and Treatment

We received the human retinal Müller cell line (MIO-M1) from Dr. G. Astrid Limb (UCL Institute of Ophthalmology, London, United Kingdom) [27]. MIO-M1 cells were continuously cultivated in Dulbecco’s Modified Eagle’s Medium (Gibco; Thermo Fisher Scientific, Inc., Waltham, MA) containing 25 mM glucose, 10% (*v*/*v*) fetal bovine serum (Biowest, Nuaille, France), 100 U/mL of penicillin, and 100 µg/mL of streptomycin at 37 °C in a humidified 5% CO_2_ atmosphere until sub-confluence. After passage culture, the cells were treated for up to 72 h with ER stress inducer Tm (0.5 µg/mL) or Tg (1 µM) in the presence or absence of nobiletin (4–64 µM) or nobiletin congeners (64 µM).

### 2.3. Cell Viability Analysis Using alamarBlue^®^

Cell viability was evaluated using alamarBlue^®^ reagent (Invitrogen; Thermo Fisher Scientific, Inc., Waltham, MA) in accordance with the manufacturer’s instructions. Briefly, MIO-M1 cells were counted and seeded at 1 × 10^5^ cells/well in a 96-well plate and cultured for 48 h with or without Tm (0.5 µg/mL) or Tg (1 µM). Following treatment, the cells were stained with alamarBlue^®^ dye, the fluorescence intensity of the incorporated reagent was measured at 530 nm (excitation) and 590 nm (emission), and the values were calculated in relation to those of the controls considered as 100%.

### 2.4. Cell Viability Analysis Using Trypan Blue

We conducted cell viability analysis using trypan blue. Briefly, MIO-M1 cells were counted and seeded at 1 × 10^5^ cells/well in a 6-well plate and cultured for 48 h with or without the tested compounds. Following treatment, the cells were subjected to trypsin digestion followed by centrifugation at 400× *g* for 5 min to collect the cells. Collected cells were re-suspended with Ca^2+^- and Mg^2+^-free Dulbecco’s phosphate buffered saline (PBS) (−), and then, we mixed even amounts (100 µL) of cell-suspending and 0.4% trypan blue solutions (FUJIFILM Wako Pure Chemical Co.). We analyzed blinded specimens using a hemocytometer and determined the quantity of cell death as the number of blue cells/total cells. The specimens were then unblinded, and the numerical values from four independent tests were averaged. We obtained digital images using an OLYMPUS CKX53 microscope (Olympus Co., Tokyo Japan) at a magnification of 40×.

### 2.5. DNA Fragmentation Analysis

For DNA fragmentation analysis, we cultured MIO-M1 cells in a 100 mm dish until sub-confluence. Following treatment with Tm (0.5 µg/mL) or Tg (1 µM), adherent cells were harvested by scraping in lysis buffer containing 0.5% (*v*/*v*) Triton^®^ X-100, 10 mM ethylenediaminetetraacetic acid (EDTA), and 10 mM Tris-HCl (pH 7.4), followed by centrifugation at 20400× *g* for 7 min to extract total DNA. The supernatants including DNA fragments were incubated with 5 µg/mL of RNase, DNase-free (Roche Applied Science, Mannheim, Germany) for 1 h at 37 °C and then treated with 4 µg/mL of proteinase K for 30 min at 50 °C to deactivate RNase activity. The purified DNA solution was incubated with 1M NaCl and 50% isopropanol overnight at −20 °C. After centrifugation at 20,400× *g* for 20 min, the isolated DNA fragments were re-dissolved with loading buffer containing 0.05% (*w*/*v*) bromophenol blue and 6.7% (*w*/*v*) sucrose and then subjected to electrophoresis with 2% (*w*/*v*) agarose. After electrophoresis, we visualized the DNA fragments by staining them with ethidium bromide.

### 2.6. Western Blot Analysis

MIO-M1 cells were numbered and seeded at 1 × 10^5^ cells/well in a 6-well plate. Following treatment with Tm (0.5 µg/mL) in the presence or absence of nobiletin (64 µM), cell fractions were collected using Blue Loading Buffer (Cell Signaling Technology, Inc.) containing 62.5 mM Tris-HCl (pH 6.8), 2% (*w*/*v*) sodium dodecyl sulfate (SDS), 10% glycerol, 0.01% (*w*/*v*) bromophenol blue, and 41.6 mM dithiothreitol. The isolated cellular proteins were subjected to Western blot analysis using specific rabbit antibodies against CHOP (L63F7) (1:2000 dilution), BiP (1:2000 dilution), and caspase-3 (1:1000 dilution) and specific mouse antibodies against β-actin (1:10,000 dilution). We used the β-actin antibody as a loading control to check equal loading of protein on each lane. We detected immunoreactive CHOP and BiP using enhanced chemiluminescence (ECL) Western Blotting Detection Reagent (GE Healthcare, Buckinghamshire, UK) after the isolated cellular proteins were complexed with horseradish-peroxidase-conjugated anti-rabbit immunoglobulin G (IgG). For the detection of GAPDH and Siah-1, cell fractions were prepared using radio-immunoprecipitation assay (RIPA) buffer containing 1% Nonidet P-40, 0.5% sodium deoxycholate, 0.1% SDS, 200 µM phenylmethylsulphonyl fluoride (PMSF), 100 µM sodium orthovanadate, and proteinase inhibitor cocktail solution. Quantitative analyses were conducted using Image J (National Institutes of Health (NIH), Bethesda, MD), as previously reported [16]. We quantitatively determined the comparative volume of the immunoreactive proteins by densitometric scanning using Amersham Imager 680 (Cytiva, Tokyo, Japan).

### 2.7. Immunofluorescence Analysis

MIO-M1 cells were numbered and seeded at 4 × 10^4^ cells/well insert of a cell disk (Sumitomo Bakelite Co., Ltd., Tokyo, Japan) with a 24-well plate. After treatment with Tm or Tg in the presence or absence of nobiletin (64 µM), we fixed the cells in a solution of 4% paraformaldehyde for 15 min at room temperature and washed them twice with PBS(−). We infiltrated the cells with ice-cold methanol for 10 min, blocked them with 1% goat serum in PBS(−), and then incubated them at 4 °C with a specific antibody against GAPDH (1:300 dilution). The cells were then washed twice with PBS(−), followed by 1 h of incubation with anti-goat secondary antibody conjugated to Alexa Fluor 488 (1:100 dilution) at room temperature. After nuclear staining with propidium iodide, cell disks were washed extensively in PBS(−) and mounted on glass slides using ProLong^®^ Diamond Antifade Mountant (Life Technologies; Thermo Fisher Scientific, Inc.). We investigated GAPDH nuclear accumulation under blind condition using a fluorescent microscope (Olympus). We determined the percentage of nuclear GAPDH-positive cells that translocated GAPDH from the cytosol to the nucleus based on the detection principle, as previously reported [28]. To quantify the proportion of GAPDH-positive nuclei in retinal Müller cells, the numbers of propidium iodide (PI)-positive and PI/GAPDH double-labeled nuclei in five different fields/specimens were counted. We estimated nuclear-GAPDH-positive cells (%) as nuclear-GAPDH-positive cells/total nuclear staining cells. The specimens were then unblinded, and the numerical values from four independent tests were averaged.

### 2.8. Lactate Dehydrogenase (LDH) Analysis

MIO-M1 cells were counted and seeded at 5 × 10^3^ cells/well in a 96-well plate. Following treatment with Tm in the presence or absence of nobiletin (4–64 µM) or nobiletin congeners (64 µM), cell death was quantitatively detected using cytotoxicity analysis based on the release of lactate dehydrogenase (LDH) using the LDH-Cytotoxic Test Wako kit (FUJIFILM Wako Pure Chemical Co.).

### 2.9. siRNA Transfection

MIO-M1 cells were transfected with siRNAs targeting Siah-1 using Lipofectamine^®^ RNAiMAX (Invitrogen; Thermo Fisher Scientific, Inc.). The transfection was conducted at 37 °C in a humidified 5% CO_2_ atmosphere. After seeding MIO-M1 cells at 5 × 10^4^ cells/well in a 6-well plate, transfection was conducted by mixing RNA interference (RNAi) (25 pmol) with 9 µL of Lipofectamine RNAiMAX in a final volume of 250 µL of serum-free Opti-Minimal Essential Medium (Opti-MEM) without antibiotics. The sequences for siRNA were human Siah-1 (sense 5′-GCAUCAGCAUAAGUCCAUUtt-3′, antisense 5′-AAUGGACUUAUGCUGAUGCat-3′). The cells were divided into three groups and transfected with Siah-1 siRNA, negative control (scramble)-siRNA, and blank control. Transfections were performed in triplicate, and the experiment was repeated four times.

### 2.10. Real-Time Reverse Transcription–Quantitative Polymerase Chain Reaction (RT-qPCR)

MIO-M1 cells were numbered and seeded at 1 × 10^5^ cells/dish in a 60 mm dish. Following treatment with Tm in the presence or absence of nobiletin (4–16 µM), we isolated total RNA using the ISOGEN reagent (Nippon Gene, Toyama, Japan), and first-strand complementary DNA (cDNA) was synthesized from 500 ng of RNA using the PrimeScript^®^ RT reagent kit (Takara Bio, Inc., Shiga, Japan) in accordance with the manufacturer’s instructions. The densities of the synthesized cDNA solution were finally 50 ng/µL. The cDNA solution was applied in qPCR to analyze VEGF or PEDF expression levels using an SYBR green PCR master mix (Takara Bio, Inc., Shiga, Japan). The primers for qPCR were human VEGF-A (forward 5′-TCACAGGTACAGGGATGAGGACAC-3′, reverse 5′-CAAAGCACAGCAATGTCCTGAAG-3′), human PEDF (forward 5′-CCCATGATGTCGGACCCTAA-3′, reverse 5′-GATGATACTCATGCTTCCGGTCAA-3′), and human β-actin (forward 5′-TGGCACCCAGCACAATGAA-3′, reverse 5′-CTAAGTCATAGTCCGCCTAGAAGCA-3′). We performed PCR according to the following procedure: denaturation at 95 °C for 30 s, followed by 40 cycles of extension at 95 °C for 5 s and 64 °C for 34 s. Threshold marks were automatically regulated to intersect amplification marks in the linearity portion of the amplification curves, and the cycles to threshold (Ct) were saved by themselves. Data were canonicalized to the expression of the intrinsic control, β-actin. We performed qPCR using the Applied Biosystem 7500 Real Time PCR System (Life Technologies; Thermo Fisher Scientific, Inc.).

### 2.11. Measurement of Polymethoxylated Flavones in Ocular Tissues after Intraperitoneal Administration in Rats

Ten male Wistar rats aged 6–7 weeks and weighing approximately 200 g (Sankyo Labo Service Co. Ltd., Tokyo, Japan) were used for these experiments. All rats were group-housed in suspended wire-bottomed cages with food and water administered ad libitum, in a 12 h light/12 h dark schedule. All animal experiments (Ethical Code Number: 19-006; date of approval: 10 June 2019) described in this study were handled in accordance with the Association for Research in Vision and Ophthalmology Statement for the Use of Animals in Ophthalmic and Vision Research. After the feeding period, food was withheld for 12 h. Fermented *Citrus reticulata* (ponkan) fruit squeezed draff containing nobiletin and 4′-demethlated nobiletin dissolved in 50% ethanol (400 mg/mL) was then administered intraperitoneally at a dosage of 1000 mg squeezed draff powder per kg body weight. Rats were sacrificed 30 min after administration by the process of CO_2_ euthanasia, and then blood and whole-eye samples were collected. To examine the distribution of nobiletin and 4′-demethylated nobiletin in ocular tissue, the whole eyes were then dissected carefully into four parts (cornea, sclera with choroid, retina, and lens). The plasma and corresponding tissues of both eyes from the animal were pooled, with the tissues being used to measure the polymethoxylated flavones in the different eye tissues after administration of fermented *Citrus reticulata* (ponkan) fruit squeezed draff by high-pressure liquid chromatography.

### 2.12. Statistical Analysis

Data are expressed as the mean ± standard deviation, and we analyzed the data using Student’s *t*-test. *p* < 0.05 was considered to indicate a statistically significant difference for all analysis.

## 3. Results

### 3.1. ER-Stress-Induced Müller Cell Apoptotic Death

ER stress can be chemically induced by tunicamycin (Tm) or thapsigargin (Tg); therefore, we examined the actions of ER stress inducers Tm and Tg on Müller cell survival. As shown in Figure 1, the results of alamarBlue^®^ analysis showed that both Tm and Tg decrease cell viability. To examine whether ER stress induces apoptotic cell death in Müller cells, we performed Western blot analysis. MIO-M1 cells were treated with Tm or Tg for up to 72 h and subsequently analyzed for cleaved caspase-3, which is a key marker of future late-stage apoptosis. As shown in Figure 2A, the expression of cleaved caspase-3 was observed after treatment with Tm or Tg in MIO-M1 cells. In addition, we detected the expression of CHOP, which is known as growth arrest and DNA damaged-inducible gene 153 (GADD153), induced in the apoptotic pathway (Figure 2B). These results indicated that ER stress induces apoptotic cell death in Müller cells. ER stress activates several ER-stress-related protein expressions, such as BiP, which belong to the unfolded protein response pathway. We then examined the effect of Tm or Tg on ER-stress-related proteins. As shown Figure 2B, both Tm and Tg elevated the expression of BiP in addition to CHOP, demonstrating that the activation of the unfolded protein response signaling pathway is caused by these ER stress inducers in Müller cells.

### 3.2. Nuclear Translocation of GAPDH in ER-Stress-Induced Müller Cell Death

Nuclear translocation of GAPDH is associated with apoptotic Müller cell death under hyperglycemic conditions [29] and is involved in the development of DR [30]. We next examined subcellular translocation of GAPDH in ER-stress-induced Müller cells. In normal Müller cells, GAPDH was mainly localized in the cell cytoplasm. Treatment with Tm caused significant accumulation of nuclear GAPDH (Figure 3, middle row). Tg also induced nuclear translocation of GAPDH compared to that in control cells (Figure 3, lower row).

### 3.3. Significant Reduction in ER-Stress-Induced Cell Death in Siah-1-Knockdown Müller Cells

For nuclear localization of GAPDH, GAPDH needs to bind Siah-1 with the nuclear localization signal, forming a complex and subsequently allowing for the transport of GAPDH to the nucleus [31]. To confirm whether nuclear localization of GAPDH is definitely associated with ER-stress-induced cell death, we established Siah-1-knockdown Müller cells using Siah-1-directed siRNA. We examined the induction potential of cell death by Tm in Siah-1-knockdown Müller cells. As shown in Figure 4A, Siah-1 siRNA (25 pmol) significantly decreased Siah-1 mRNA expression at 48 h by 68.0 ± 7.5% compared to that by the non-RNAi transfection control (*p* < 0.001). However, Siah-1 mRNA levels were less affected by the transfection of scramble siRNA in MIO-M1 cells. In scramble siRNA transfection cells, the percentage of cell death induced by Tm was 61.2 ± 3.0%. There was no significant difference in the percentage of cell death between un-transfected and scramble siRNA-transfected cells in Tm-treated MIO-M1 cells. The percentage of induced cell death significantly diminished by 46.6 ± 3.3% in Siah-1-knockdown Müller cells compared to un-transfected cells (*p* < 0.05) (Figure 4B). From these results, we suggested that inhibition of GAPDH nuclear translocation is partially associated with a reduction in Müller cell death.

### 3.4. Effect of Nobiletin on ER-Stress-Induced Müller Cell Death

Next, we examined whether nobiletin exerts a protective action against ER-stress-induced Müller cell death. The morphologic images showed the protective action of nobiletin against Tm-induced Müller cell death (Figure 5A). As shown in Figure 5B, cell viability analysis using trypan blue demonstrated that Tm increased Müller cell death by 2.5 ± 0.4-fold and the percentage of cell death induced by Tm significantly diminished from 56.9 ± 7.7% to 38.3 ± 4.9% after co-treatment with Tm and nobiletin (*p* < 0.01). LDH analysis demonstrated a dose-dependent action of nobiletin on the prevention of Tm-induced Müller cell damage (Figure 5C). In contrast, nobiletin did not inhibit Tg-induced Müller cell death (Figure 6).

### 3.5. Regulation of ER-Stress-Related Proteins and Cleaved Caspase-3 Expression by Nobiletin in Müller Cells

We further investigated the action of nobiletin on ER-stress-related proteins, BiP and CHOP, in Tm-treated Müller cells. Nobiletin was found to significantly attenuate the Tm-elevated BiP expression at a concentration of 64 µM (*p* < 0.05) (Figure 7A). In contrast, nobiletin significantly augmented the Tm-induced CHOP expression (*p* < 0.05) (Figure 7B). Western blot analysis also demonstrated that nobiletin suppresses the Tm-induced upregulation of cleaved caspase-3 in Müller cells (Figure 7C).

### 3.6. Nobiletin Inhibits ER-Stress-Induced Nuclear Translocation of GAPDH in Müller Cells

To clarify the regulation of GAPDH nuclear translocation by nobiletin, we investigated the action of nobiletin on GAPDH nuclear accumulation under ER stress conditions. As shown in Figure 8, the percentage of MIO-M1 cells positive for nuclear GAPDH decreased significantly by 18.3 ± 7.4% or 48.0 ± 7.6% in nobiletin–Tm-co-treated cells (*p* < 0.05) following 24 or 48 h of treatment, respectively, compared to 58.7 ± 26.3% (24 h) or 72.9 ± 15.1% (48 h) in Tm-treated cells. We investigated the action of nobiletin on Siah-1 and GAPDH protein expression in Müller cells, but nobiletin showed no effect on Siah-1 and GAPDH protein expression in Tm-treated MIO-M1 cells (data not shown). In contrast, nobiletin did not show inhibitory action against the Tg-induced nuclear translocation of GAPDH (Figure 9).

### 3.7. Effect of Nobiletin on ER-Stress-Mediated VEGF and PEDF Expression in Müller Cells

We investigated the action of nobiletin on the expression levels of VEGF and PEDF in Müller cells. As shown in Figure 10, in MIO-M1 cells, Tm was found to significantly increase VEGF mRNA expression but decrease PEDF mRNA expression at 24 h after treatment. Nobiletin showed no action on the Tm-induced VEGF expression level. However, nobiletin significantly augmented PEDF expression in Tm-treated cells (*p* < 0.05). These results suggested that nobiletin protects BRB integrity through the augmentation of PEDF expression, leading to an improvement in the VEGF/PEDF ratio.

### 3.8. Effect of Nobiletin Congeners on ER-Stress-Induced Müller Cell Death

Following the previously reported protocol [24,25,26], we synthesized nobiletin congeners, and the structures of nobiletin (compound 1) and the congeners (compounds 2–25) used in this study are shown in Figure 11. We examined the effect of nobiletin congeners on Tm-induced retinal Müller cell death by LDH analysis. As shown in Figure 11, we could find nobiletin congeners, compound 6, compound 18, and compound 24, which showed significant inhibition of Tm-induced cell damage as well as nobiletin in retinal Müller cells (compound 6, *p* < 0.01; compound 18, *p* < 0.05; compound 24, p < 0.05).

### 3.9. Ocular Distribution of Nobiletin and 4′-Demethylated Nobiletin

Finally, we performed experiments to evaluate the ocular distribution of nobiletin and the demethylated congener 4′-demethylated nobiletin using fermented *Citrus reticulata* (ponkan) fruit squeezed draff, which abundantly contains these flavones [32]. As shown in Table 1, the nobiletin concentration in the retina was found to be markedly higher than the levels in plasma after intraperitoneal administration in rats, suggesting that nobiletin can translocate to the retina from circulating blood. However, the retina/plasma ratio of the concentration of 4′-demethylated nobiletin was lower than that of nobiletin.

## 4. Discussion

Researchers have shown interest in the usage of Müller cells as a novel therapeutic target in the early stages of several retinal degenerative diseases, including DR [33]. The contact of Müller glia with the subretinal space and vitreous make them accessible to both subretinally and intravitreally injected drugs, which have benefits in several therapeutic applications. In addition, Müller glia are involved in the composition and secretion of neurotrophic factors, growth factors, and cytokines [34], which leads to the support of the survival of photoreceptors and other retinal neurons. In contrast, in DR, high-glucose and other diabetic insults, including ER stress, trigger Müller cell death. Therefore, pharmacological intervention for inhibiting Müller cell death may present new opportunities to prevent retinal injury in diabetes. We demonstrated that nobiletin, a polymethoxylated flavone from citrus explants, has inhibitory potential in ER-stress-induced Müller cell death. In addition, Müller cell death caused by ER stress was found to be closely associated with GAPDH nuclear translocation. Nobiletin could inhibit ER-stress-induced GAPDH nuclear translocation, which resulted in the protection of Müller cells.

In this study, we demonstrated that ER stress induces apoptotic cell death in retinal Müller cells, through experiments for the detection of DNA fragmentation, cleaved caspase-3, and CHOP expression, which are key markers of apoptosis. Previous reports have suggested that nuclear accumulation of GAPDH plays a role in apoptotic Müller cell death under high-glucose conditions [29] and is involved in the development of DR [30]. Although the functions of GAPDH within the nucleus involved in retinal cell death have not fully been identified, there is a report that GAPDH in the nucleus is likely to be associated with the regulation of p53 vitalization, a key protein strongly involved in the derivation of some types of cell death [35]. Yego et al. [36] demonstrated that hyperglycemia acts as a stimulus for GAPDH nuclear accumulation in Müller cells by the hyperglycemia-stimulated activation of a caspase-1/interleukin-1β signal pathway. In this study, ER stress was also found to be one of the inducing stimuli for GAPDH nuclear translocation in Müller cells. In addition, because the ER-stress-induced Müller cell death was significantly inhibited in Siah-1-knockdown Müller cells, Siah-1 protein is necessary for ER-stress-induced GAPDH nuclear accumulation, which results in Müller cell death. Although nobiletin did not affect GAPDH and Siah-1 protein expression, it could suppress Tm-induced GAPDH nuclear translocation. In contrast, nobiletin did not suppress Tg-induced GAPDH nuclear localization and could not inhibit Müller cell death. These results suggest that prevention of Müller cell death by nobiletin is associated with the inhibition of GAPDH nuclear translocation. However, the reason for the inability of nobiletin to protect against Tg-induced cell death remains to be fully elucidated. Tg is shown to bind to the ATP-binding pocket, cause cooperative inhibition of sarco-/endoplasmic reticulum Ca^2+^-ATPase, and induce ER stress in various cells. In addition, Tg-induced apoptotic cell death is reported to be involved with the Tg-associated mitochondrial protein, prohibitin-1, through the elevation of [Ca^2+^]_i_ and ROS production. Karthikeyan et al. [37] demonstrated that epigallocatechin gallate, a flavonoid, could inhibit Tg-mediated apoptosis by inhibiting the expression of prohibitin-1 in mouse retinal pigment epithelial cells. Therefore, we need to examine the action of nobiletin on Tg-associated proteins, including prohibitin-1 in Müller cells. BiP is an important ER chaperone, and its expression is triggered by ER stress and accumulation of misfolded proteins [38]. Several studies have demonstrated that BiP upregulation participates in apoptosis induction [39,40]. The suppression of BiP expression in Tm-treated Müller cells by nobiletin suggested that in addition to the inhibition of GAPDH nuclear translocation, nobiletin may protect Müller cells from ER stress by reducing the accumulation of unfolded and misfolded proteins. However, nobiletin did not affect Tm-induced CHOP expression; therefore, CHOP may have little involvement in the inhibitory action on ER-stress-induced apoptosis by nobiletin.

The pharmacological actions of PMFs are directly related to their chemical structure and depend on the number and positions of hydroxyl groups and methoxy groups [41,42]. For example, we reported that the hydroxylation of 4′-methoxy group at the B-ring on the flavone structure leads to the augmentation of inhibitory action against MMP-9 activity in Müller cells [16]. In this study, there was no significant difference between nobiletin congeners with the 4′-hydroxy group at the B-ring (compound 3, compound 6, and compound 8) and the corresponding compound with the 4′-methoxy group at the B-ring (nobiletin, compound 2, and compound 4); therefore, we speculated that 4′-demethylation of the methoxy group at the B-ring does not necessarily lead to the augmentation of inhibitory potential against ER-stress-induced Müller cell death by PMFs. In addition, the retina/plasma ratio of concentration of 4′-demethylated nobiletin was found to be lower than that of nobiletin. These results suggested that hydroxylation of the 4′-methoxygroup at the B-ring may lead to the attenuation of BRB permeability by PMFs. Liu et al. [43] assessed the BRB permeability of polyphenols, including that of PMFs, in an in vitro BRB model consisting of the human retinal pigment epithelial cell line and demonstrated that BRB permeability of flavonoids is improved by methylation at the side chain of the structure. These results may provide support to our supplemental results related to BRB permeability of PMFs in vivo.

In conclusion, Müller cells have unique characteristics, such as maintenance of BRB integrity and stem cell potential, that make them suitable targets for the prevention and treatment of DR. In this study, we demonstrated that nobiletin prevents ER-stress-induced Müller cell death by inhibiting the translocation of GAPDH to the nucleus. Therefore, therapies targeted toward preventing GAPDH nuclear localization in retinal Müller cells should help in arresting the development and also the progression of DR. We need further research to clarify weather ER stress actually induces Müller cell death through GAPDH nuclear translocation in animal models. We will examine the preventive action on Müller cell death in diabetic retina by nobiletin. In addition, we will examine nobiletin-augmented PEDF expression in Müller cells, which may lead to the protection of BRB integrity through improvement in the VEGF/PEDF ratio. Müller cells have been shown to become mired in dysfunction in the early stages of DR; therefore, we consider that adequate modulation of Müller cell responses by intake of nobiletin may lead to a promising prevention therapy for DR. Prospects of this research include conjugating the nobiletin molecule with a nanocarrier for effective and sustainable delivery to the degenerating retina.

## Figures and Tables

**Figure 1 cells-10-00669-f001:**
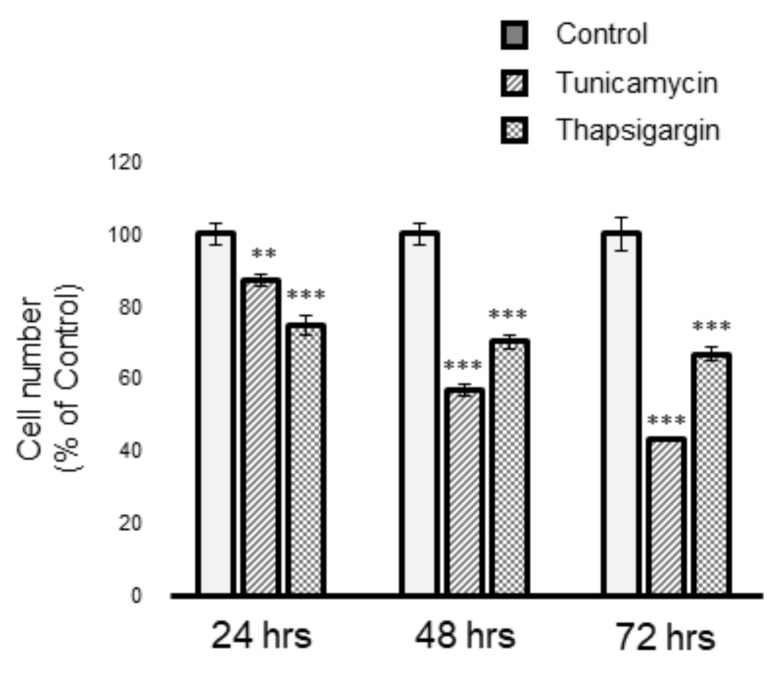
Endoplasmic reticulum (ER)-stress-induced Müller cell death. Cell viability was evaluated based on alamarBlue analysis, following treatment with tunicamycin (0.5 µg/mL) or thapsigargin (1 µM) for up to 72 h. Data are shown as the mean ± SD (n = 4) of four experimental trials. ** and *** Significantly different from untreated cells at each time (*p* < 0.01 and 0.001, respectively).

**Figure 2 cells-10-00669-f002:**
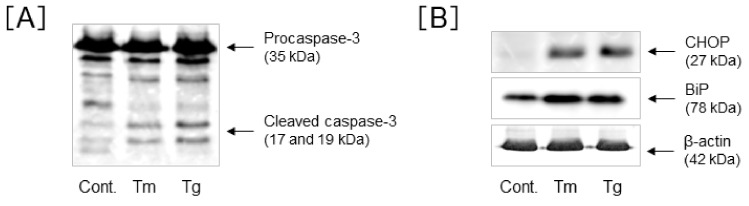
ER-stress-induced Müller apoptotic cell death. MIO-M1 cells were treated with tunicamycin (Tm) (0.5 µg/mL) or thapsigargin (Tg) (1 µM) for up to 72 h. After 72 h treatment, cellular proteins were collected, and then we conducted Western blot analysis for cleaved caspase-3 (**A**). Cellular proteins were collected after 24 h treatment, and then we conducted Western blot analysis for ER-stress-related proteins BiP and C/EBP homologous protein (CHOP) (**B**).

**Figure 3 cells-10-00669-f003:**
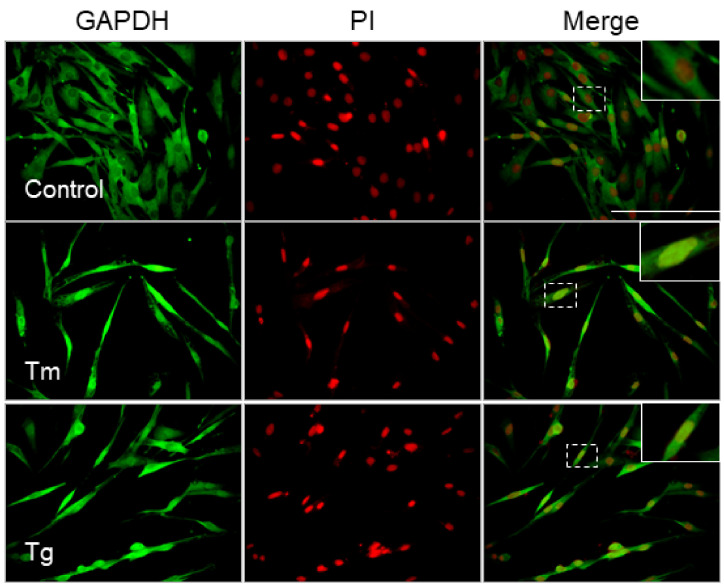
Nuclear translocation of glyceraldehyde 3-phosphate dehydrogenase (GAPDH) in ER-stress-induced Müller cell death. MIO-M1 cells were treated with tunicamycin (Tm) (0.5 µg/mL) or thapsigargin (Tg) (1 µM) for up to 48 h. The representative images of immunofluorescence staining of glyceraldehyde 3-phosphate dehydrogenase (GAPDH; green) and propidium iodide (PI; red) and the merging of GAPDH- and PI-stained images are shown at 48 h after treatment. The magnified illustration of intracellular localization of GAPDH is shown in the top corner of the untreated merged image. Scale bars, 100 µm.

**Figure 4 cells-10-00669-f004:**
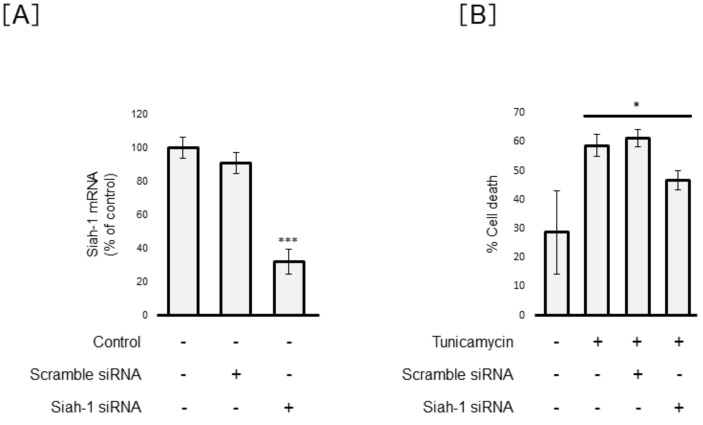
Significant reduction in ER-stress-induced cell death in Siah-1-knockdown Müller cells. MIO-M1 cells were transfected with 25 pmol of scrambled small interfering RNA (siRNA) or Siah-1 siRNA. (**A**) At 48 h post-transfection, we confirmed Siah-1 mRNA levels using RT-qPCR analysis for evaluating the transfection efficiency. Data are shown as the mean ± SD (n = 4) of four experimental trials. *** Significantly different from un-transfected cells (*p* < 0.001). (**B**) We evaluated cell viability using trypan blue. Data are shown as the mean ± SD (n = 4) of four experimental trials. * Significantly different from tunicamycin-treated cells (*p* < 0.05).

**Figure 5 cells-10-00669-f005:**
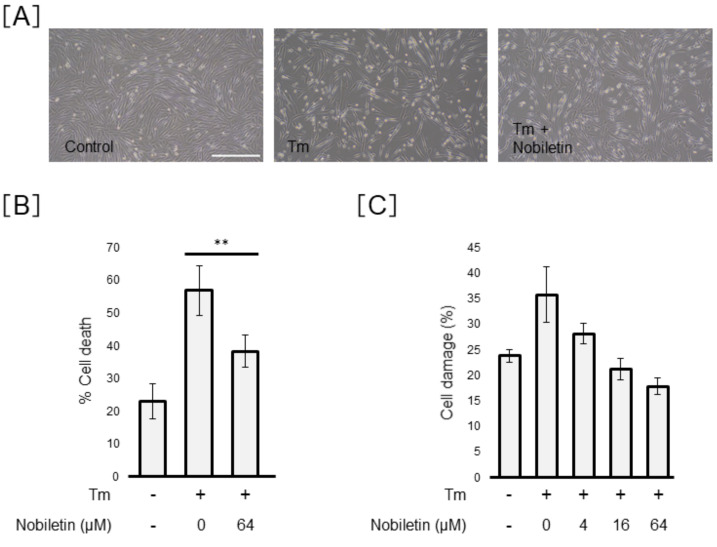
Effect of nobiletin on tunicamycin-induced Müller cell death. MIO-M1 cells were treated with tunicamycin (Tm) (0.5 µg/mL) in the presence or absence of nobiletin (4–64 µM) for 48 h. (**A**) Morphological images were shown to evaluate cell viability. Scale bars, 400 µm. (**B**) We evaluated cell viability using trypan blue. Data are shown as the mean ± SD (n = 4) of four experimental trials. ** Significantly different from tunicamycin-treated cells (*p* < 0.01). (**C**) We performed lactate dehydrogenase (LDH) analysis to confirm the dose-dependent inhibitory action of nobiletin on Tm-induced cell damage.

**Figure 6 cells-10-00669-f006:**
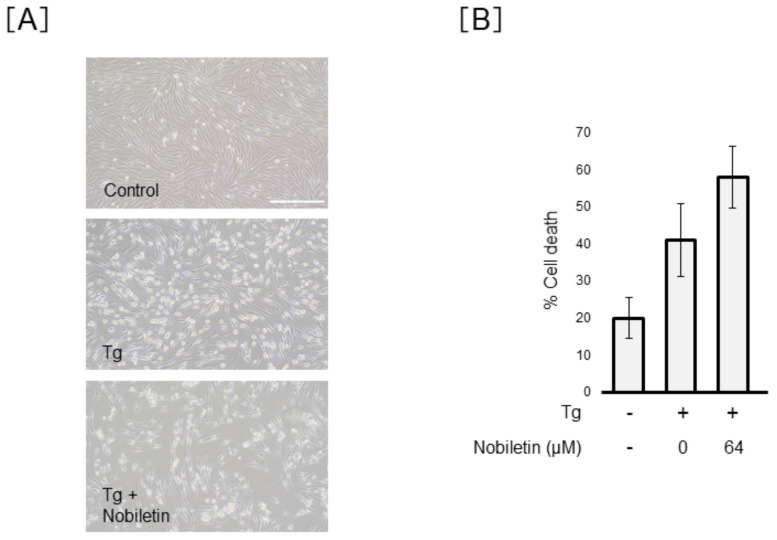
Effect of nobiletin on thapsigargin-induced Müller cell death. MIO-M1 cells were treated with thapsigargin (Tg) (1 µM) in the presence or absence of nobiletin for 48 h. Cell viability was evaluated based on morphological changes (**A**) and trypan blue viability analysis (**B**). Data are shown as the mean ± SD of four independent experiments. Scale bars, 400 µm.

**Figure 7 cells-10-00669-f007:**
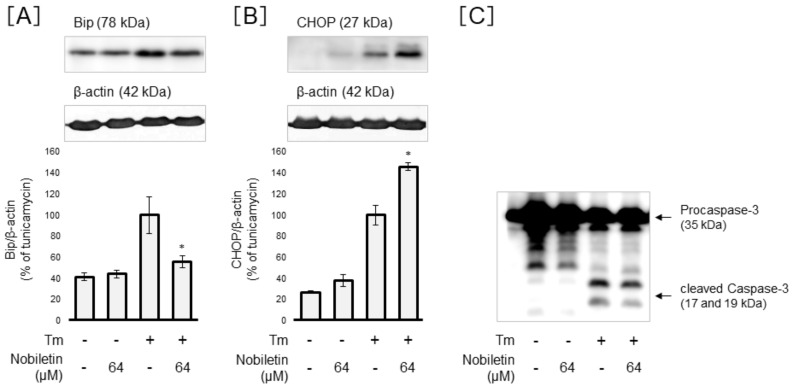
Regulation of ER-stress-related proteins and cleaved caspase-3 expression by nobiletin in Müller cells. MIO-M1 cells were treated with tunicamycin (Tm) (0.5 µg/mL) in the presence or absence of nobiletin (64 µM) for up to 72 h. Cellular proteins were collected after 24 h treatment and then subjected to Western blot analysis for ER-stress-related proteins, BiP (**A**) and CHOP (**B**). After 72 h treatment, cellular proteins were collected, and then we conducted Western blot analysis for cleaved caspase-3 (**C**). Data are shown as the mean ± SD (n = 4) of four experimental trials. * Significantly different from Tm-treated cells (*p* < 0.05).

**Figure 8 cells-10-00669-f008:**
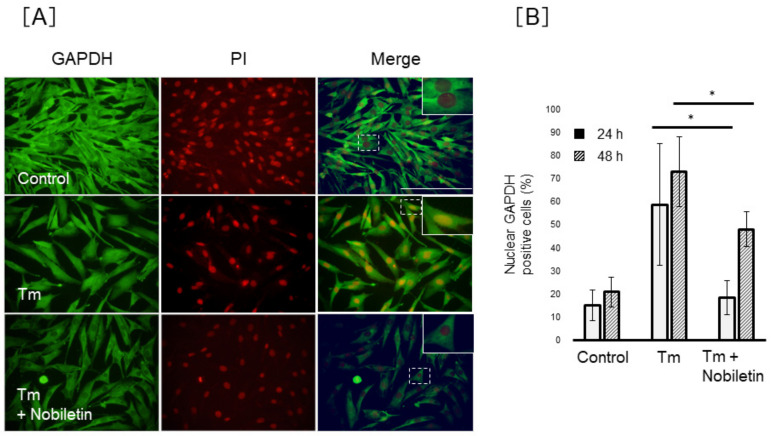
Effect of nobiletin on tunicamycin-induced nuclear translocation of GAPDH in Müller cells. MIO-M1 cells were treated with tunicamycin (Tm) (0.5 µg/mL) in the presence or absence of nobiletin (64 µM) for 48 h. (**A**) Representative pictures of immunofluorescence staining of glyceraldehyde 3-phosphate dehydrogenase (GAPDH; green) and propidium iodide (PI; red) and the merging of GAPDH- and PI-stained images are shown. The magnified illustration of intracellular localization of GAPDH is shown in the top corner of the untreated merged image. Scale bars, 100 µm. (**B**) The percentage of positive cells for nuclear GAPDH was estimated in five different fields/samples. Data are shown as the mean ± SD (n = 4) of four experimental trials. * Significantly different from Tm-treated cells at each time (*p* < 0.05).

**Figure 9 cells-10-00669-f009:**
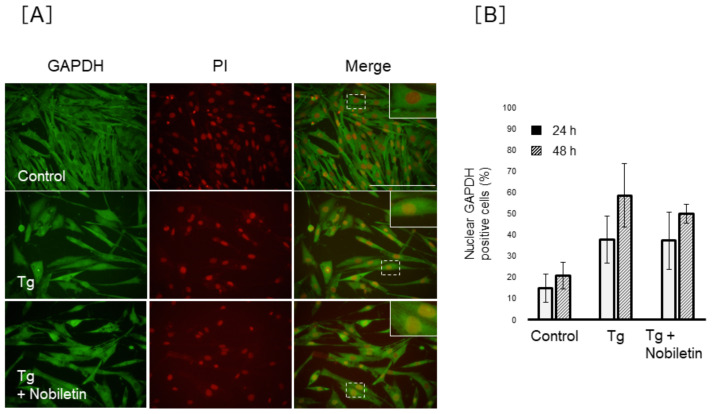
Effect of nobiletin on thapsigargin-induced nuclear translocation of GAPDH in Müller cells. MIO-M1 cells were treated with thapsigargin (Tg) (1 µM) in the presence or absence of nobiletin for 48 h. (**A**) The representative pictures of immunofluorescence staining of GAPDH (green) and PI (red) and also the merging of GAPDH- and PI-stained images are shown. The magnified illustration of intracellular localization of GAPDH is shown in top corner of the untreated merged image. Scale bars, 100 µm. (**B**) The percentage of cells that were positive for nuclear GAPDH was estimated in five different fields/samples. Data are shown as the mean ± SD (n = 4) of four independent experiments.

**Figure 10 cells-10-00669-f010:**
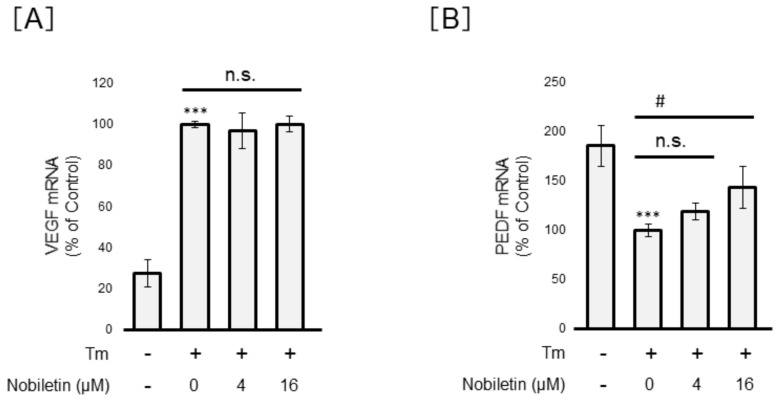
Effect of nobiletin on ER-stress-mediated vascular endothelial growth factor (VEGF) and pigment-epithelium-derived factor (PEDF) expression in Müller cells. MIO-M1 cells were treated with tunicamycin (Tm) (0.5 µg/mL) in the presence or absence of nobiletin (4–16 µM) for 24 h. Total RNA were collected and then subjected to RT-qPCR for vascular endothelial growth factor (VEGF) (**A**) or pigment-epithelium-derived factor (PEDF) (**B**). The relative gene expression of VEGF or PEDF is expressed by considering the expression of untreated cells as 100%. Data are shown as the mean ± SD (n = 4) of four experimental trials. *** Significantly different from untreated cells (*p* < 0.001); # significantly different from Tm-treated cells (*p* < 0.05); n.s., not significant.

**Figure 11 cells-10-00669-f011:**
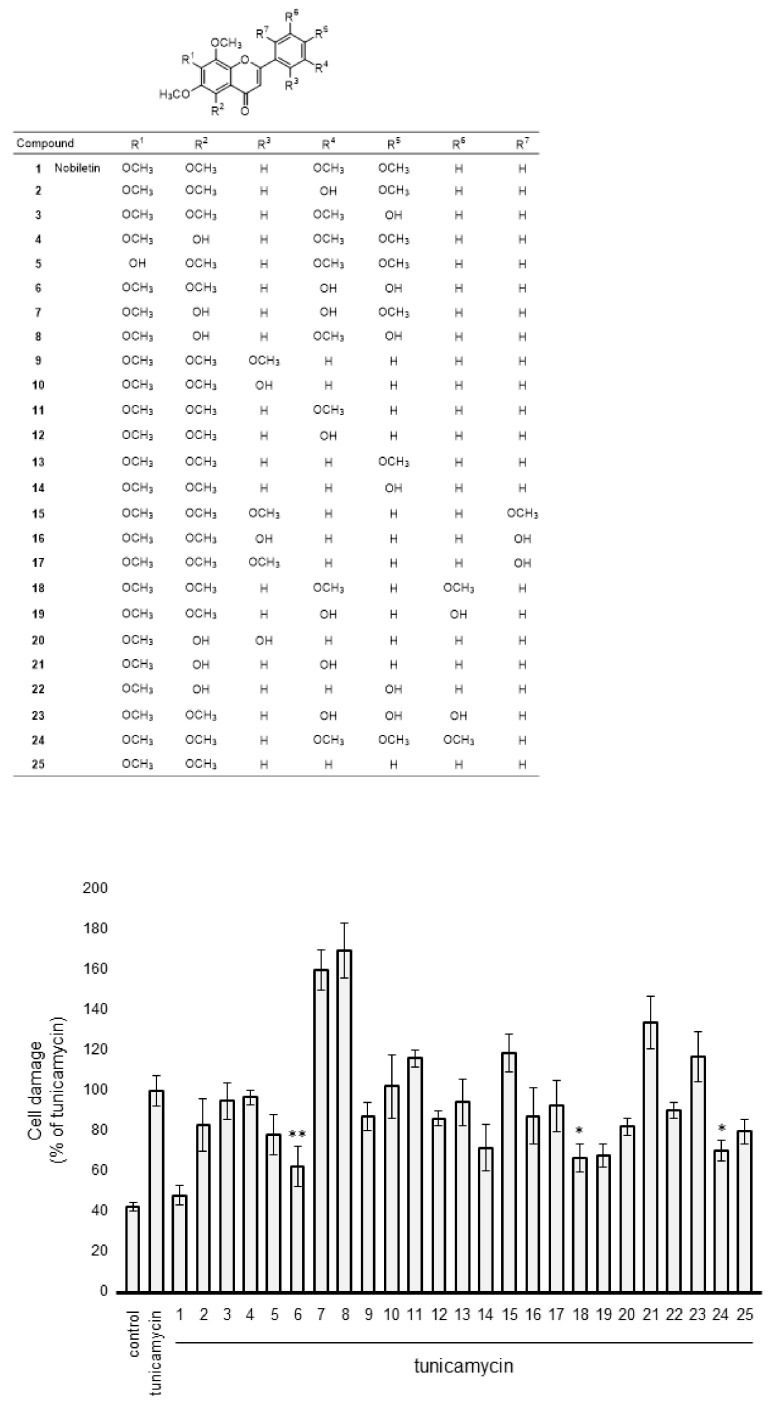
Evaluation of protective action by nobiletin congeners on ER-stress-induced Müller cell death. MIO-M1 cells were treated with tunicamycin (Tm) (0.5 µg/mL) in the presence or absence of the congeners (compounds 2–25) for 48 h. Cell viability was evaluated based on LDH analysis. Data are shown as the mean ± SD (n = 4) of four independent experiments. *and **, Significantly different from untreated cells (*p* < 0.05 and 0.01, respectively).

**Table 1 cells-10-00669-t001:** The ocular distribution of nobiletin and 4′-demethylated nobiletin at 30 min post-intraperitoneal administration of 1000 mg/kg of fermented *Citrus reticulata* (ponkan) fruit squeezed draff in Wistar rats.

Ocular Tissues or Plasma	Nobiletin (µg/g tissue)	4′-Demethylated Nobiletin (µg/g tissue)
Cornea	12.8 ± 7.8	14.8 ± 8.9
Lens	03.7 ± 1.4	03.5 ± 1.3
Retina	26.9 ± 10.0	23.2 ± 11.2
Sclera with choroid	21.3 ± 19.6	23.2 ± 26.3
Plasma (µg/mL)	15.5 ± 5.4	18.5 ± 6.6

## Data Availability

Data are contained within the article.

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
