# Peer review of "Regulation of Endothelium-Reticulum-Stress-Mediated Apoptotic Cell Death by a Polymethoxylated Flavone, Nobiletin, Through the Inhibition of Nuclear Translocation of Glyceraldehyde 3-Phosphate Dehydrogenase in Retinal Müller Cells"

_cells, 2021, doi:10.3390/cells10030669_

Round 1
Reviewer 1 Report
Dear Authors,
Thank you very much for giving me an opportunity to read your exciting manuscript. It is essential to search, test, and finally, implement new chemicals or new therapeutic approaches to deal with health issues, which is diabetic retinopathy connected with the loss of sight in affected people. Interestingly, the authors investigated interesting science areas on the crossroads between cellular apoptosis, ER stress and cell signalling events. I find it beneficial for this manuscript to improve several things to clarify and raise its informative advantages. Please be kind to see a list of questions listed below:
1], in my opinion, it is preferable to write “[4,9,10]” (instead of[4,9-10]).
2] Introduction – authors mention hypoxia as a “potent stimulus”, but there were no hypoxic conditions used in presented experiments. It is also, in my opinion, worth mentioning in an additional few word link between hypoxia and expression/activation of MMP9 as well as VEGF. Also, hypoxia-induced oxidative stress and epithelium integrity is worth considering...
3] 2.4 “PBS (-)” It is clear for me that it means –Ca –Mg free, however for a person with no background in cell biology techniques it might require some explanation in text.
4]. 2.5 – DNA fragmentation analysis: In my humble opinion, those results require some additional improvement. Is it a fact that there were differences in DNA amount per lane? Why authors did not use more frequently used methods like comet assay and others? Those results, in my opinion, are difficult to estimate/calculate/quantify and therefore other should be applied.
All text: continuous misspelling Beta-actin
5] Gapdh nuclear translocation:
- was there applied a pattern recognition software to determine a signal intensity [like ImageJ or others]?
- How it is determined that cellular morphology is of no influence on readout/ analysis? For example cell around division might show a little cytoplasm area and therefore be qualified as positive. Maybe an additional cytoplasm stain would be beneficial?
6] Figure 3 (and others). The reviewer finds a notation system over bar graphs a bit confusing. Why in F3.b 24h Tg bar is not statistically significant compared to the untreated group? There were similar unclear [my opinion]: Should not be marked, in F8.a first bar as (*) instead if second? Maybe some horizontal whiskers [like in 8b] could be a solution?
7] Fig.6. I find no B-actin pictures attached.
8] discussion- I would possibly re-consider the use of words “famous” and “irritation”
9] supplementary data S4- did authors use any statistics to present this data?
10] Did authors considered the use of repeated measures ANOVA when applicable?
Author Response
We are grateful to reviewer 1 for the critical comments and useful suggestions that have helped us to improve our paper considerably. As indicated in attached file, we have taken all these comments and suggestions into account in the revised version of our paper. We modified the manuscript based on the reviewer comments. We changed the color of modification to yellow in revised manuscript.

Reviewer 2 Report
In the present study, manuscript cells-1104460, entitled, Regulation of endothelium reticulum stress-mediated apoptotic cell death by polymethoxylated flavone, nobiletin through inhibition of nuclear translocation of glyceraldehyde 3-phosphate dehydrogenase in retinal Müller cells by Yoshiki Miyata the authors show the recent advance in role of Müller cells in endothelium reticulum stress-mediated apoptotic cell death and the potential therapeutic targets reflected according to the literature. The authors demonstrated that nobiletin, a polymethoxylated flavone from Citrus explants, has inhibitory potential on ER stress-induced
Müller cell death. In addition, Müller cell death caused by ER stress was found to be
closely associated with GAPDH nuclear translocation. The rationale behind this review is correct and well-founded but need to improve our manuscript. I haven’t objections to publish this manuscript in Cell after addressing few minor points:
- Please make the corrections marked in yellow color the changes we made on the revised version of the manuscript.
- There is a spelling error in Introduction about “occluding”
- In the results section: All the figures have a low resolution. Please improve resolution in all images, specially: Figure 1A, figure 3A, figure 5A and figure7A. The authors should change it and please provide a clear image showing the nuclear translocation by GADPH. In these figures is not possible to see the expression inside (activated) and outside the nucleus (Also it need an additional image with orthogonal view and colabelling for another marker that indicate the nuclear translocation.) Moreover, these figures require scale bar.
- Please indicate in the text, the explanation of figure 2A and 2B. Moreover, change the figure legend because is indicated BiP and CHOP in figure 2C and cleaved caspase 3 in figure 2D.
- Figure 4: In Figure 4A, no statistically significant differences are seen between groups. Please remove P value from result section. In figure 4B there is a mismatch because the text put p< 0.01 and the figure legend put p < 0.05
- In Figure 5C no statistically significant differences between groups. Please remove p< 0.01 and the figure legend.
- Discussion:
With this results of the paper it cannot be inferred that the hydroxylation of the 4'methoxygroup prevents the rupture of BRB because I haven't this table in retina/plasma. In the supplemental material not provide tables and figures. Only have figure legends.
Overall, the rationale is justified and the study design is generally sound other than a few points as below. The conclusions are supported by the results presented.

Author Response
We are grateful to reviewer 2 for the critical comments and useful suggestions that have helped us to improve our paper considerably. As indicated in attached file, we have taken all these comments and suggestions into account in the revised version of our paper. We modified the manuscript based on the reviewer comments. We changed the color of modification to yellow in revised manuscript.

Reviewer 3 Report
The manuscript by Miyata et al demonstrates that ER stress induces cell death in Muller glial cells and that treatment with nobiletin prevents cell death.
The manuscript presents the limited amount of data, all about the ER stress-induced cell death in Muller glial cells. Moreover, the data is quite superficial and limited to in vitro studies using a cell line. Once the authors hypothesize that this might have a role in Diabetic retinopathy, demonstrating it in an animal model of Diabetic retinopathy would increase the value of the presented findings.
A major concern is the considerable lack of detail about how the experiments were performed regarding the number of independent experiments, the n, time points, doses used, and origin of the material used. Although some information is also provided in the figure legends, this should be clearly stated on material and method sections.
The data described in the supplementary figures should be described in the result section and not in the discussion.
The authors should update their references once no recent ones are provided.
Minor:
Introduction
“ER stress was caused by hypoxia, a potent stimulus of angiogenesis and inflammation in retinal endothelial cells [17].” The sentence is confusing and should be rewritten.
Material and methods
Once the is no space limitation all the supplementary material and methods should be incorporated in the main manuscript
There is no mention about the number of independent experiments and “n” used
2.1
- small interfering RNA (siRNA) sequences should be used to the manuscript.
- Please correct β-actin
- “chemicals from Sigma Aldrich (St. Louis, MO).” All the chemicals used should be indicated in the main manuscript
2.3 “with or without the tested compounds” which compounds were used and what the concentration of the same
2.5 “Following treatment,” to which treatment are the authors referring to?
2.6. Western blot analysis
- How the total protein was quantified and how much protein was used to perform the western blot should be mentioned
The source of nobiletin should be clearly stated.
Results
Once the treatments caused cell death it is difficult to assess GAPDH percentage of cells (Fig 3).
What is the reason for the huge SD in Figure 4B on the control situation?
Author Response
We are grateful to reviewer 3 for the critical comments and useful suggestions that have helped us to improve our paper considerably. As indicated in attached file, we have taken all these comments and suggestions into account in the revised version of our paper. We modified the manuscript based on the reviewer comments. We changed the color of modification to yellow in revised manuscript.

Round 2
Reviewer 1 Report
Dear Authors,
Thank you very much for your kind response and for your modifications that include my remarks.
I do consider my questions fully answered, except a few I listed below.
1) DNA fragmentation analysis.
There were little to no changes to the presented data. As I suggested before, I would strongly advise using more, in my opinion, established methods like comet assay or tunnel assay whereas those presented in fig. 1a or 1b are unevenly applied into a gel. You could see faint smear also in negative samples which make it all impossible to compare. I would suggest reconsidering using those particular data, also due to possible "technical" bias caused by low sample number [just one per treatment].
2) GADPH transition in fig 3a is difficult to be clearly determined due to change in cell shape and cytoplasm volume caused by treatment. Some picture postprocessing with nucleus/cytoplasm ration would be beneficial. Another concern is a fact that only a picture from a confocal microscope with a Z stack] can 100% deliver a clear evidence of changes in nucleolar staining.
3] Although I welcome improvement in data presentation, figures 4b, 5b and 8b remain unclear, only from a visual perspective. It might help to rearrange an order of the bars or join relevant bars like in fig. 10 or in the work of Shimada A. et al (2014) from "Endocrinology Diabetes and Obesity".
Kind regards,
Author Response
We are grateful to reviewer 1 for the critical comments and useful suggestions that have helped us to improve our paper considerably. As indicated in the responses that follow, we have taken all these comments and suggestions into account in the revised version of our paper. We modified the manuscript based on the reviewer comments. We changed the color of modification to yellow in revised manuscript. Thank you very much for your critical indication for our future research.
Comment #1.
The result of DNA fragmentation analysis in Figure 2 is not quantitative, not accepted.
Response.
According to your indication, we removed the result of DNA fragmentation analysis in Figure 2. Aside from result of DNA fragmentation analysis, in this study, we examined the expression of cleaved Caspase-3 and CHOP which is key markers involved in apoptosis induction. Taking together these results, we concluded that ER stress induces apoptotic cell death in retinal Müller cells. We changed the description of results related to Figure 2 in Result section.
Comment #2.
In Figure 3A, it is difficult to be clearly distinguish area of nucleus and cytoplasm due to change in cell shape.
Response.
Although we did not use confocal microscope with a Z stack to obtain the results shown in Figure 3A, we think that the immunocytochemical analysis clearly demonstrates that ER stress inducer, tunicamycin or thapsigargin could induce GAPDH translocation. On the other hand, reviewer 1 has question about the accuracy related to percentage of cells positive for nuclear GAPDH , which is calculated from the results of Figure 3A. According to your indication, we removed these quantitative results as shown in Figure 3B.
Comment #3.
In Figures 4B, 5B and 8B, the author should rearrange an order of the bars like Figure 10
Response.
According to your suggestion, we modified the description of bars in Figure 4B, 5B and 8B.

Reviewer 3 Report
The authors did not improve the scientific content of the manuscript, with new data, for example by adding the analysis of diabetic retinas (in/ex vivo) However, they were able to improve the description of their experimental approach. This increased the quality of the manuscript and will allow the reader to better understand what was done.
Figure 7 and 8: it should be clear from which area are the insets. Please add a (dotted) box to the figure.
Author Response
We are grateful to reviewer 3 for the critical comments and useful suggestions that have helped us to improve our paper considerably. As indicated in the responses that follow, we have taken all these comments and suggestions into account in the revised version of our paper. We modified the manuscript based on the reviewer comments. We changed the color of modification to yellow in revised manuscript. Thank you very much for your critical indication for our future research.
Comment #1.
It should be clear from which are the insets in Figure 3, 8 and 9
Response.
According to your indication, we made dotted box to each Figure.
